# Novel Therapeutic Approaches for Alzheimer’s Disease: An Updated Review

**DOI:** 10.3390/ijms22158208

**Published:** 2021-07-30

**Authors:** Tien-Wei Yu, Hsien-Yuan Lane, Chieh-Hsin Lin

**Affiliations:** 1Department of Psychiatry, Kaohsiung Chang Gung Memorial Hospital, Chang Gung University College of Medicine, Kaohsiung 83301, Taiwan; mp9596@cgmh.org.tw; 2Department of Psychiatry and Brain Disease Research Center, China Medical University Hospital, Taichung 40402, Taiwan; 3Graduate Institute of Biomedical Sciences, China Medical University, Taichung 40402, Taiwan; 4Department of Psychology, College of Medical and Health Sciences, Asia University, Taichung 41354, Taiwan; 5School of Medicine, Chang Gung University, Taoyuan 333, Taiwan

**Keywords:** Alzheimer’s disease, amyloid, tau, NMDA, neuroinflammation, neuroprotection, brain stimulation, rTMS, tDCS, precision medicine

## Abstract

Alzheimer’s disease (AD) is a progressive neurodegenerative disease and accounts for most cases of dementia. The prevalence of AD has increased in the current rapidly aging society and contributes to a heavy burden on families and society. Despite the profound impact of AD, current treatments are unable to achieve satisfactory therapeutic effects or stop the progression of the disease. Finding novel treatments for AD has become urgent. In this paper, we reviewed novel therapeutic approaches in five categories: anti-amyloid therapy, anti-tau therapy, anti-neuroinflammatory therapy, neuroprotective agents including N-methyl-D-aspartate (NMDA) receptor modulators, and brain stimulation. The trend of therapeutic development is shifting from a single pathological target to a more complex mechanism, such as the neuroinflammatory and neurodegenerative processes. While drug repositioning may accelerate pharmacological development, non-pharmacological interventions, especially repetitive transcranial magnetic stimulation (rTMS) and transcranial direct current stimulation (tDCS), also have the potential for clinical application. In the future, it is possible for physicians to choose appropriate interventions individually on the basis of precision medicine.

## 1. Introduction

Alzheimer’s disease (AD) is a progressive neurodegenerative disease and the leading cause of dementia in the elderly [1]. Worldwide, around 50 million people have dementia, and 50–70% of cases are attributed to AD [2,3]. Both the prevalence and incidence of AD increase with age. Globally, the population aged 65 years or older is expected to increase from 9.3% in 2020 to around 16.0% in 2050 [4]. In the United States, the prevalence of AD is approximately 3% in people aged 65–74, 17% in people aged 75–84, and 32% in people aged 85 or older [5]. The incidence of AD doubles every 10 years in those aged older than 60 [6]. Currently, about 5.8 million American adults suffer from AD, and the number is predicted to reach nearly 14 million by 2050 [7].

AD causes functional disability in the elderly. Typical characteristics of AD are progressive memory loss and functional impairment. AD not only impacts the individual but also their families and society. In 2016, the Global Burden of Disease classification system listed AD as the fourth highest disease for premature death and the sixth most burdensome disease [8]. Patients with AD develop behavioral and psychological symptoms of dementia (BPSD), including delusions, misperceptions, mood disorders, and behavioral disturbances [9]. The presentation of BPSD increases the burden on caregivers [10]. Patients with AD or other dementia types require about 170 h of informal care per month, which is a twofold increase compared to those without dementia [11]. The heavy care burden of AD leads to physical, psychological, and financial impacts on both families and society.

Despite the profound and chronic effects of AD, current treatments are unable to achieve satisfactory therapeutic effects or stop disease progression. Today, only five drugs have been approved by the FDA for AD treatment: donepezil, rivastigmine, galantamine, tacrine, and memantine. The first four drugs are acetylcholinesterase inhibitors (AChEIs), while the last one is an N-methyl-D-aspartate receptor (NMDAR) antagonist [12]. American and European guidelines list AChEIs as first-line pharmacotherapies for mild to moderate AD. However, AChEIs only show modest efficacy on cognitive deficits and non-significant efficacy on functional capacity in mild to moderate AD [13]. Memantine shows very limited efficacy on cognitive symptoms without functional improvement [14]. Finding novel treatments for AD has become urgent.

Understanding AD pathogenesis may guide the development of novel treatments. Traditionally, the pathological hallmarks of AD include two misfolded proteins: β-amyloid (Aβ) and tau. Aβ deposition links to tau accumulation [15]. Tau accumulation is associated with glucose hypometabolism, brain atrophy, and neurodegeneration [16,17]. Some biological mechanisms also drive protein aggregation, including carriage of the apolipoprotein E type 4 allele (APOE4), neuroinflammation, sleep disturbance, and autophagy dysfunction [18,19].

This article aims to review novel therapeutic approaches to AD, including pharmacological interventions (Table 1) and non-pharmacological interventions (Table 2).

## 2. Novel Therapeutic Approach

### 2.1. Anti-Amyloid Therapy

Amyloid plaques are composed of Aβ peptides in the extracellular space. Aβ is derived from the amyloid precursor protein (APP), a transmembrane protein. β-secretase and γ-secretase cleave the APP and generate pathological Aβ [138]. Accumulation of Aβ results in neurotoxicity [139,140]. Reducing the accumulation of Aβ has become a therapeutic purpose of AD [141]. Anti-amyloid therapy consists of three strategies: secretase inhibitors, Aβ aggregation inhibitors, and Aβ immunotherapy.

#### 2.1.1. Secretase Inhibitors

Secretase inhibitors target the catalytic activities of β-secretase and γ-secretase, which is the rate-limiting step in Aβ production. These strategies have been studied for the past two decades. Inhibitors of β-secretase (BACE)1 decreased the Aβ levels in AD patients’ cerebrospinal fluid (CSF) [20,142]. Several BACE1 inhibitors have reached phase III clinical trials, such as verubecestat [21], atabecestat [22], lanabecestat [23], LY3202626 [24], and umibecestat [25], but these drugs have failed due to a lack of efficacy or worse cognitive function in patients with mild cognitive impairment (MCI), and mild to moderate AD [143]. The last BACE inhibitor, elenbecestat, was discontinued in phase III trials because it showed an unfavorable risk/benefit ratio in early AD [26,27]. The inhibition and modulation of γ-secretase were both therapeutic straggles. Double-blind randomized controlled trials (RCTs) of the γ-secretase inhibitors, including semagacestat [28] and avagacestat [29], were discontinued. Cognitive deterioration was noted in patients with MCI and mild to moderate AD. The γ-secretase modulator tarenflurbil also worsened cognition in patients with mild AD [30]. Therefore, the role of secretase inhibitors remains under debate [144,145].

#### 2.1.2. Aβ aggregation Inhibitors

Several natural compounds have Aβ aggregation inhibitory properties [146]. However, multiple obstacles blocked these compounds from entering into clinical use. First, some compounds have poor permeability through the blood–brain barrier (BBB). Second, these compounds are small molecules and produce insufficient steric effects to disrupt Aβ aggregation. Third, the protein–protein binding regions are relatively featureless to small molecules without specific pockets or grooves [147,148].

One strategy is to target the chaperones in the brain, such as metals. Disrupting the interaction between Aβ peptides and metals provides a barrier against Aβ oligomerization. Metal protein attenuating compounds (MPACs) chelate copper and zinc ions and inhibit Aβ aggregation [149,150,151]. One such example is clioquinol (PBT1), a hydroxyquinoline ionophore. In patients with MCI to moderate AD, clioquinol showed no significant improvement in cognition or clinical global impression between the active treatment and placebo groups. Subgroup analysis showed that clioquinol treatment rescued cognitive decline in the more severely affected patients (The Alzheimer’s Disease Assessment Scale–Cognitive Subscale (ADAS-cog) ≥ 25). The adverse effect of visual impairment was reported in the treatment group [31]. Second-generation clioquinol (PBT2) had appeared to be safe and well tolerated in people with mild AD. However, the double-blind RCTs of PBT2 demonstrated no overall significant effect on cognition or function in treating MCI and mild to moderate AD [32,33].

Recently, advanced biophysical and structural biology experimental approaches have been used to investigate chemical features and identify potential compounds. Some compounds with binding epitopes or planar hydrophobic structures increase Aβ aggregation inhibitory activities, including tanshinone and uncarinic acid C. A few compounds, namely, epigallocatechin gallate (EGCG) and resveratrol, interact with the toxicity determinants of Aβ, the N-terminus and β1-turn regions. Epigallocatechin gallate (EGCG), oleuropein aglicone (OleA), and quercetin are potential therapeutic compounds for AD [147].

#### 2.1.3. Aβ Immunotherapy

Aβ immunotherapy actively or passively decreases the Aβ burden. The active anti-Aβ vaccine AN1792 was first tested in humans, but the trial was discontinued because meningoencephalitis was developed in 6% of immunized AD patients [152]. Furthermore, although four second-generation active Aβ vaccines reached phase II trials: ACI-24, CAD106, UB-311, and ABVac40, none of them were proven to be clinically beneficial in treating AD at that time [34]. Passive immunotherapy promotes Aβ clearance by targeting neurotoxic Aβ oligomers [153]. Three humanized monoclonal antibodies underwent phase III trials, including BAN2401, gantenerumab, and aducanumab. Studies involving these agents were confirmed to engage amyloid oligomers and decrease downstream tau levels. BAN2401 has demonstrated modest cognitive and functional efficacy in APOE4 carriers with MCI to mild AD [35]. Gantenerumab has shown no clinical efficacy thus far in phase III trials in prodromal to mild AD [36]. Two double-blind phase III RCTs of aducanumab were terminated halfway through in March 2019. The futility determination was based on the low chance of therapeutic efficacy for AD. However, the company gathered additional data and announced that the final result of aducanumab showed positive treatment effects [37,38]. In June 2021, the U.S. Food and Drug Administration (FDA) approved aducanumab to treat AD patients [154]. The argument about this approval seems to be continued. Next-generation oral-form small-molecule agents targeting Aβ oligomers, such as CT1812, PQ912, and ALZ-801, are also in development [35].

### 2.2. Anti-tau Therapy

The tau protein is associated with microtubules and stabilizes microtubules in axons and dendrites. Tau undergoes the process of post-translational modifications, especially hyper-phosphorylation [155,156]. Hyper-phosphorylated tau proteins accumulate and form intracellular neurofibrillary tangles (NFTs). NFTs induce inflammatory responses and cause neurotoxicity. Unlike Aβ, the development of tau pathology correlates with the severity of cognitive deficit in AD [157]. The strategies for anti-tau therapy include phosphatase modifiers, kinase inhibitors, tau aggregation inhibitors, microtubule stabilizers, and tau immunotherapy.

#### 2.2.1. Phosphatase Modifiers

Phosphatase modifiers decrease phosphorylation by activating phosphatases, such as protein phosphatase 2A (PP2A) [158]. Sodium selenate is a PP2A activator and an essential molecule in neurological functions [159]. Sodium selenite deficiency was related to oxidative damage and cognitive impairment [160]. A phase II trial on sodium selenate did not find any change in cognitive performance in mild to moderate AD [39]. A supranutritional supplement of sodium selenate increased selenium uptake in the CNS, but the clinical efficacy was minor for AD [40].

#### 2.2.2. Kinase Inhibitors

Kinase inhibitors decrease post-translational modifications and limit the hyper-phosphorylation of tau. The degree of phosphorylation is related to the activity of protein kinases: cyclin-dependent-like kinase 5 (CDK5) [161] and glycogen synthase kinase-3β (GSK3β) [162]. Selective inhibitors of CDK5 have been reported in cancer therapy, including roscovitine [163] and flavopiridol [164]. In animal models of AD, roscovitine prevented tau phosphorylation, while flavopiridol reduced memory decline [41,42]. None of these agents have reached clinical trials in AD. Two types of GSK3β inhibitors, tideglusib and lithium, have been researched in AD. Tideglusib showed no clinical benefit in phase II trials of mild to moderate AD, and its short-term administration resulted in an adverse effect of a reversible transaminase increase [43]. Lithium, a mood stabilizer, was identified as a GSK3β inhibitor. One double-blind RCT revealed that a microdose of lithium prevented cognitive decline in AD patients [44]. The participants received a 15-month lithium treatment, with a daily dose of 300 mg. Meta-analyses concluded that lithium inhibited the progression of cognitive decline in AD patients, with a moderate effect size [45,46]. Whether lithium is effective in treating AD needs further verification.

#### 2.2.3. Tau Aggregation Inhibitors

Methylene blue (MB) is a synthetic phenothiazine dye and the earliest tau aggregation inhibitor. MB blocks interactions between tau molecules and disrupts polymerization in vitro [165,166]. In a phase II double-blind RCT, 50-week administration of MB in which the participants received 138 mg of MB treatment showed a cognitive benefit in mild to moderate AD [47]. Methylthioninium chloride (LMTX), the MB derivative, failed to improve cognitive or functional performance in a phase III double-blind RCT in mild to moderate AD [48]. An advanced study demonstrated that MB inhibited tau fibril formation but accelerated the formation of neurotoxic tau oligomers [167]. Therefore, the role of MB remains ambivalent in AD therapy.

Curcumin is a coloring agent and food additive. Curcumin inhibits tau aggregation by decreasing β-sheet formation in tau and disintegrating tau oligomers in vitro [168]. Several phase II double-blind RCTs of curcumin displayed no clinical or biomarker improvement after a 6-month treatment in AD patients [169,170]. The failure of previous studies of curcumin was attributed to its low bioavailability [171]. In the cognitively healthy elderly, a bioavailability-improved formulation of curcumin administration improved working memory in an acute and short-term course (<4 weeks) [49,50]. However, a long-term course of curcumin treatment did not delay cognitive decline [172]. A recent systematic meta-analysis indicated that curcumin treatment worsened cognitive performance in AD patients [173].

#### 2.2.4. Microtubule Stabilizers

Epothilone D (EpoD) is an anti-fungal agent and a microtubule stabilizer. Epothilone D induces tubulin’s polymerization into microtubules and enhances microtubule bundling in vitro [174]. In animal studies, EpoD rescued working and spatial memory deficits in aged tau transgenic mice [175,176,177], but the phase I trial of EpoD failed due to intolerable adverse effects [51].

NAP (davunetide), an activity-dependent neuroprotective protein (ADNP) derivative, protects microtubules from katanin disruption in vitro [178,179]. In a phase II double-blind RCT, NAP showed cognitive and functional improvement in MCI, when MCI patients received a 12-week intranasal NAP administration [52,53]. The clinical effect of NAP has not been researched in AD patients yet [180].

TPI-287 (abeotaxane) is a synthetic taxane derivative for central nerve system (CNS) malignancy or metastasis treatment [181,182]. A phase I double-blind RCT of TPI-287 showed less decline in Mini-Mental State Examination (MMSE) scores in the treated group compared to placebo in mild to moderate AD. Three serious adverse events (15%) with anaphylactoid reactions were reported [54]. In addition to EpoD, NAP, and TPI-287, the development of a peptide with the taxol-binding pocket of β-tubulin has become another innovative strategy [183].

#### 2.2.5. Tau Immunotherapy

Active tau vaccines have been developed to trigger antibodies against tau proteins. Two tau vaccines have reached clinical trials: AADvac1 and ACI-35 [184]. The antibodies from AADvac1 target the microtubule-binding region of tau, decrease tau aggregation, and promote tau clearance [185]. In a phase I double-blind RCT in mild to moderate AD, almost all the patients receiving the AADvac1 injection (29/30) showed an IgG immune response within 12 weeks [186]. No case of meningoencephalitis or vasogenic edema was reported at a 72-week follow-up assessment [187]. One phase II double-blind RCT of AADvac1 was performed to evaluate its clinical efficacy in patients with mild AD [55]. ACI-35 is a liposome-based vaccine against phosphorylated tau. In animal studies, ACI-35 induced a rapid immune response and decreased phosphorylated tau in tau transgenic mice within 12 weeks [188]. One phase I double-blind RCT is underway in patients with mild to moderate AD to assess the tolerability and safety of the ACI-35 vaccine [56]. One novel tau vaccine, Aβ 3–10-keyhole limpet hemocyanin (KLH), reduced the phosphorylated tau level and improved cognitive functions in animal studies [57].

Passive immunotherapy is being developed for tau pathology. Several agents have achieved clinical trials for AD: BIIB092, ABBV-8E12, RO7105705, BIIB076, LY3303560, UCB0107, and JNJ-63733657 [189]. Three such agents are humanized lgG4 monoclonal antibodies BIIB092, ABBV-8E12, and RO7105705. Gosuranemab (BIIB092) was safe and well tolerated in healthy participants [190]. One large phase II double-blind RCT of BIIB092 is ongoing in patients with early AD [58]. ABBV-8E12 showed an acceptable safety profile in a phase I study [191]. Phase II double-blind RCTs of ABBV-8E12 are being continued with regard to the efficacy of treating patients with early AD [59,60]. Semorinemab (RO7105705) showed a fair safety profile in healthy individuals. Two phase II double-blind RCTs of RO7105705 are ongoing in prodromal to mild AD [61] and moderate AD [62].

Two agents belong to humanized lgG1 monoclonal antibodies: BIIB076 and LY3303560. BIIB076 was both safe and tolerable in healthy participants and MCI patients [63]. An advanced clinical trial of BIIB076 is not yet available. LY3303560 appeared to be tolerable in healthy individuals and AD patients [192,193]. One phase II triple-blind RCT of LY3303560 has ended in early AD, but the efficacy is currently not available [64]. JNJ-63733657, a monoclonal antibody, completed two phase I trials in healthy participants and patients with AD [194,195]. In early AD, the phase II double-blind RCT of the efficacy of JNJ-63733657 is still being studied [65]. UCB0107, a humanized version of antibody D, is undergoing a phase I investigator-blind RCT in a healthy population [66,67].

### 2.3. Anti-neuroinflammatory Therapy

Neuroinflammation contributes to the progression of AD and correlates with the severity of the disease [196]. Anti-neuroinflammatory strategies include microglia modulators, astrocyte modulators, insulin resistance management, and microbiome therapy.

#### 2.3.1. Microglia Modulators

Microglial activation is recognized as a hallmark of neuroinflammation. Microglia interact with Aβ and the tau protein in the pathogenesis of AD [197,198]. Glial activation is associated with the signaling pathways of apolipoprotein E (ApoE), thus triggering the receptor expressed on myeloid cells 2 (TREM2), Toll-like receptor (TLR), and colony-stimulating factor-1 receptor (CSF1R) (Figure 1, panel A) [199].

Mutations of ApoE and TREM2 are considered strong risk factors of AD. The ApoE-TREM2 pathway shares similar mechanisms in regulating Aβ pathology in AD [200]. APOE is a primary cholesterol carrier and identified as a ligand for human TREM2 in microglia. The interaction increases TREM2-mediated phagocytosis of apoptotic neurons [201,202]. In an AD mouse model, increased TREM2 expression led to improved memory performance in 5xFAD mice [203]. The deficiency of TREM2 decreased plaque deposition during the early stage of AD but enhanced amyloid-β pathology in the advanced stage [204]. No agent targeting ApoE or TREM2 has reached clinical trials for AD treatment.

Multiple TLR pathways respond to the accumulation of Aβ and induce neuronal injuries in AD pathogenesis, especially TLR4 and TLR2. The TLR4 pathway interacts with NLRP3 inflammasomes and sustains neuroinflammation [205]. Furthermore, the TLR4 pathway is activated by lipopolysaccharide (LPS) and induces memory impairment in animal models of AD [206]. Several TLR4 inhibitors improved cognitive deficits in AD animal models, including thymoquinone, ethyl pyruvate, and TAK-242 [68]. TLR2 binds to Aβ and mediates the Aβ phagocytosis by microglia [207]. Dysregulation of the TLR2 pathway accelerated memory impairment in AD mice, either through inhibition [208,209,210] or activation [207]. None of these agents have reached clinical trials for AD therapy.

The CSF1R pathway drives microglial proliferation in animal models of AD. Selective CSF1R inhibitors were applied in transgenic AD mice, such as GW2580, JN-J527, and PLX3397. The efficacy of GW2580 blocked microglial proliferation and recovered the short-term memory and behavioral deficit in APP/PS1 mice [69]. Administration of JN-J527 improved tau-mediated neurodegeneration and functional impairment in P301S mice [70]. Long-term treatment of PLX3397 reversed spatial and emotional memory deficits in 5XFAD mice [71].

#### 2.3.2. Astrocyte Modulators

The astrocyte reaction impairs the clearance of Aβ at the BBB. The astrocyte reaction in AD involves several signaling pathways: the Janus kinase/signal transducer and activator of transcription 3 (JAK/STAT3), the calcineurin/nuclear factor of activated T cells (calcineurin/NFAT), the nuclear factor-kB/nod-like receptor family pyrin domain containing 3(NFκB/NLRP3), the mitogen-activated protein kinase (MAPK), and the P2Y1 purinoreceptor (P2Y1R) pathways (Figure 1, panel B) [211].

The JAK/STAT3 pathway has been activated in reactive astrocyte transgenic mouse models of AD [212]. Stattic is a selective STAT3 inhibitor, and its intraperitoneal injection rescued learning and memory impairment in 5XFAD mice [72,73]. The calcineurin/NFAT pathway promotes the production of proinflammatory cytokines [213]. FK506 (Tacrolimus) inhibited the calcineurin/NFAT pathway and improved cognitive deficit in APP/PS1 mice [214,215]. An open-label phase II study of FK506 is underway to investigate the efficacy in MCI and AD. No results have yet been published [74]. The NFκB/NLRP3 pathway is activated by Aβ and promotes the production of proinflammatory cytokines [216]. Eliminating NLRP3 reduced brain Aβ levels in AD animal models [217,218]. Inhibition of the NFκB/NLRP3 pathway is a potential treatment, but no agents have yet entered clinical trials of AD [219].

P38 MAPK, a class of MAPKs, responds to inflammatory cytokines, mediates Aβ-induced neurotoxicity, and is correlated with tau phosphorylation [220]. Several p38 MAPK inhibitors were investigated in vivo, including SB202190 and PD169316 [75]. Two highly selective p38 MAPK inhibitors were investigated in animal studies of AD: MW181 and NJK14047. MW181 blocked tau phosphorylation and rescued cognitive impairment in aged hTau mice [76]. Meanwhile, NJK14047 decreased Aβ deposits, decreased neuron death, and improved cognitive functions in 5XFAD mice [77]. The P2Y1R pathway increases the frequency of spontaneous astroglial calcium events. The process promotes downstream p38 activity and glutamate-induced neuronal death in AD mice [221,222]. Several P2Y1R inhibitors have been involved in AD studies: MRS2179 and BPTU. Treatment of P2Y1R inhibitors normalized astrocyte activity and improved cognitive deficits in APPPS1 mice [78].

#### 2.3.3. Insulin Resistance Management

AD features deficits in cerebral glucose utilization with progressive cognitive impairment [223]. The deficits in cerebral glucose utilization in human AD include insulin deficiency, insulin-like growth factor 1 (IGF-1) deficiency, and insulin resistance. Insulin resistance promotes oxidative stress, triggers inflammation, and increases tau phosphorylation and toxic Aβ levels [224].

Insulin therapy is applied when treating AD with an intranasal device. A double-blind RCT reported that intranasal insulin administration improved memory impairment in MCI and AD. The participants received intranasal regular insulin at 40 IU daily for 4 months [79]. A systematic review of RCTs indicated that patients with MCI and AD displayed improved verbal memory after insulin therapy. The patients without an APOE4 gene had more consistent cognitive benefits than the APOE4 carriers [80]. A recent RCT of intranasal insulin therapy also failed in treating MCI and AD. After 12 months of treatment, the treated group demonstrated no significant difference in cognition and function compared to the placebo group [81]. Intranasal insulin therapy is a relatively safe option of treatment without serious adverse events in the treated group [80].

Incretins are gut-derived hormones that stimulate insulin secretion, including glucagon-like peptide-1 (GLP-1) and glucose-dependent insulinotropic polypeptide (GIP). Several incretin receptor agonists showed a potential therapeutic effect in animal models of AD and Parkinson’s disease: liraglutide, lixisenatide, exendin-4, semaglutide, peptide 17, peptide 18, peptide 20, DA-JC4, and DA-CIB [82]. A double-blind RCT of liraglutide was examined in AD treatment. The 12-month liraglutide treatment delayed cognitive impairment in the treated group compared to the placebo group [83]. Another phase II double-blind RCT of liraglutide is ongoing in patients with mild AD [225].

Metformin is the first-line therapy for diabetes mellitus. In diabetic patients, metformin demonstrated a neuroprotective effect and reduced the risk of developing dementia [226,227]. One study of metformin involved non-diabetic, overweight (BMI over 25) populations with MCI. The treated group received 500–2000 mg of metformin daily. After a 12-month intervention, the treated group showed a reduction in recall memory decline compared to the placebo group [84]. One pilot crossover RCT of metformin was tested in non-diabetic and non-overweight adults with MCI and early AD. The participants were randomized to receive an 8-month metformin or a placebo intervention. The daily dose of metformin was as high as 2000 mg. The results showed that the metformin administration improved executive functions in the treated group compared to placebo [85].

Peroxisome proliferator activator receptors (PPARs) mediate the anti-inflammatory process and metabolic pathways [228]. Three isotypes of PPARs have been identified: PPAR-α, PPAR-β/δ, and PPAR-γ. Four PPAR-α agonists showed therapeutic potential in animal models of AD: WY-14643, GW7647, fenofibrate, and gemfibrozil. A phase I trial of gemfibrozil in MCI patients has been completed, and advanced clinical studies are pending [86]. Pioglitazone is a PPAR-γ agonist for treating diabetes. An open-label phase II RCT of pioglitazone showed cognitive benefits in diabetic patients with mild AD. The participants received 15–30 mg of pioglitazone daily for 6 months [87]. Two phase III quadruple-blind trials of pioglitazone in MCI patients were terminated due to a lack of efficacy without safety concerns [88,89]. The PPAR-δ agonists have been evaluated in AD mouse models [229]. A hybrid PPAR-δ and PPAR-γ agonist, T3D-959, resolved neuroinflammation in an intracerebral streptozotocin (STZ) animal model of AD [90].

#### 2.3.4. Microbiome Therapy

The composition of the gut microbiota affects the gut–brain communication and brain function by synthesizing various neurotransmitters and neuromodulators [230]. Dysbiosis of the gut microbiota leads to an overproduction of LPS in the gut, which increases permeability to the BBB [231]. Sodium oligomannate (GV-971), a marine-derived oligosaccharide, suppresses gut microbiota dysbiosis, regulates neuroinflammation, and destabilizes Aβ aggregates [232]. Phase III double-blind RCTs of sodium oligomannate showed a cognitive benefit in patients with mild to moderate AD [91]. The participants received a dose of 900 mg of sodium oligomannate for 36 weeks. The treated group showed significant improvement in ADAS-cog performance compared to the placebo group [92]. Sodium oligomannate was approved in November 2019 in China for treating mild to moderate AD [233].

### 2.4. Neuroprotective Agents

Neurodegenerative mechanisms are involved in the pathogenesis of AD. Therefore, applying neuroprotective strategies aims to delay both the AD onset and AD progression [234]. Three neuroprotective candidates are generally discussed: antiepileptic drugs, NMDAR modification, and omega 3 polyunsaturated fatty acid supplements.

#### 2.4.1. Antiepileptic Drugs

Antiepileptic drugs are considered CNS depressants and have been found to deteriorate cognitive functions. In recent studies, some agents had the potential to enhance cognitive performance in epileptic patients, including levetiracetam and gabapentin [235].

Levetiracetam exerts a therapeutic effect by targeting the synaptic vesicle 2A (SV2A) protein [236]. Levetiracetam displayed neuroprotective properties in traumatic brain injury in both animal models and clinical trials [237]. In a mouse model of AD, administration of levetiracetam decreased the Aβ load and rescued the cognitive deficit in APP/PS1 mice after a 4-week treatment [238]. In the healthy elderly, the double-blind crossover RCT of levetiracetam showed potential with regard to enhancing cognitive functions. The volunteers received a dose of 1000 mg of levetiracetam during the 5-week treatment phase. The volunteers in the levetiracetam-treated phase showed cognitive improvement but with a tendency of irritability and fatigue [239]. In patients with MCI, a multicenter double-blind phase III RCT of AGB101 (levetiracetam) is currently ongoing to evaluate its potential to slow cognitive and functional decline [93].

Gabapentin is a voltage-gated calcium channel (VGCC) inhibitor that indirectly affects the glutamate system. In cerebral ischemia-reperfusion mice, gabapentin treatment showed a neuroprotective effect and reduced neural injury in a dose-dependent manner [240]. In healthy populations, administration of a single dose of 50–400 mg of gabapentin promoted a subtle cognitive improvement [241]. In dementia patients with BPSD, preliminary evidence indicated that gabapentin treatment had possible benefits in treating AD. The gabapentin treatment with a daily dose of 200–3600 mg decreased agitation and improved cognition. The result was based on low-grade evidence [242]. A double-blind phase IV RCT of gabapentin enacarbil is continuing to investigate the therapeutic efficacy of nighttime agitation and restless leg syndrome in patients with moderate to severe AD [94].

#### 2.4.2. NMDAR Modification

Glutamate is one of the major excitatory neurotransmitters in the CNS. The N-methyl-D-aspartate receptor (NMDAR) is a subtype of the ionotropic glutamate receptor and plays a critical role in regulating synaptic plasticity, neuronal survival, learning, and memory [243]. Individuals with AD had decreased glutamate levels in CSF and fewer NMDARs in the hippocampus and frontal cortex [244]. Enhancement or modulation of NMDAR activity demonstrated therapeutic potential in early AD.

The preservative sodium benzoate enhances NMDAR activity by inhibiting D-amino acid oxidase (DAAO). D-serine, the main co-agonist of NMDARs, is metabolized by DAAO into hydroxypyruvate. Inhibition of DAAO increases the level of downstream D-serine (Figure 2) [245]. In studies of schizophrenia, sodium benzoate inhibited reactive oxygen species and had a potent neuroprotective effect [246,247,248]. Sodium benzoate was tolerated in patients with MCI and mild AD. The participants received a 24-week benzoate treatment with a dose of 250 to 750 mg per day. The treated group showed greater improvement in ADAS-Cog than the placebo group [95]. In a phase II double-blind RCT, 24-week administration of sodium benzoate in which the participants received 250–1500 mg benzoate treatments showed both altered brain activity and cognitive benefit in MCI [96]. In patients with BPSD, a 6-week benzoate treatment demonstrated a benefit in specific individuals: those with a young age, those of the female gender, those with a higher BMI, those with a significant DAAO decrease, and those with antipsychotic use [247]. In a multicenter, double-blind RCT, benzoate treatment showed cognitive benefits in women with moderate to severe AD. The treated group received 250–1500 mg of benzoate daily for 6 weeks and showed improved ADAS-Cog performance compared to the placebo group [97].

Riluzole is classified as a glutamate modulator and is used in amyotrophic lateral sclerosis therapy. Riluzole inhibits the presynaptic glutamate release indirectly and modulates the postsynaptic NMDAR activity. In animal models of early AD, the riluzole-treated group had better enhanced cognition and a reduced Aβ load compared to the placebo group in transgenic mice [249,250]. A phase II double-blind RCT of riluzole was completed to assess the cerebral metabolism and cognitive effect in mild AD. No results have yet been published [98]. Troriluzole (BHV-4157) is a riluzole derivative, and one phase II double-blind RCT of troriluzole is continuing to evaluate the cognitive change in patients with mild to moderate AD [99].

#### 2.4.3. Omega 3 Polyunsaturated Fatty Acid Supplements

Omega 3 polyunsaturated fatty acids include three subtypes: α-linolenic acid (ALA), eicosapentaenoic acid (EPA), and docosahexaenoic acid (DHA). The latter two were derived from fish oil and demonstrated an anti-inflammatory effect against cardiovascular diseases [251]. In AD mouse models, a supplement of either EPA or DHA showed a neuroprotective property and improved memory and learning [252]. In MCI patients, several controlled studies have indicated that omega 3 fatty acid supplements from 3 to 12 months significantly improved cognitive performance over the placebo [253,254]. In APOE4 carriers, phospholipid DHA dietary supplements had the potential to prevent the development of AD [255]. A phase II double-blind RCT has been ongoing to evaluate the effect of the APOE4 genotype and the cognitive efficacy of DHA supplements [101]. In mild to moderate AD, a phase III RCT of a DHA supplement was evaluated. The 18-month 2 mg DHA supplements administered daily did not rescue the cognitive and functional decline in the treated group when compared to the placebo group [100]. A recent systematic review and meta-analysis study suggested that only combined DHA and EPA supplements improved certain aspects of cognitive performance in AD patients. No consistent evidence has supported the therapeutic efficacy in short- or medium-term treatment [256]. A phase III RCT of icosapent ethyl, an ethyl ester of EPA, is ongoing to evaluate the cognitive and cerebrovascular effect in cognitively healthy adults at increased risk for AD [102].

### 2.5. Brain Stimulation

Brain stimulation is proposed as a promising non-pharmacological therapeutic option for AD [257]. In the field of AD therapy, several brain stimulation methods have been researched: deep-brain stimulation (DBS), vagus nerve stimulation (VNS), transcranial magnetic stimulation (TMS), and transcranial electrical stimulation (Figure 3) [258].

#### 2.5.1. Deep-Brain Stimulation

Deep-brain stimulation is an invasive brain stimulation technique. The surgeon implants electrodes at a targeted region of the brain and promotes electrical stimulation. The electrical stimulation is provided by an implantable pulse generator. DBS treatment is considered an advanced treatment for tremors in patients with Parkinson’s disease [259].

In 2010, a small-size phase I trial of DBS was investigated in six patients with mild AD. The DBS was placed in the fornix within the hypothalamus. After 12 months of continuous stimulation, the patients showed reduced cognitive decline and improved glucose metabolism at the temporoparietal lobe [260]. A phase II double-blind RCT of forniceal DBS was performed in mild AD patients. No significant difference in cognition or metabolism was observed between the treated and control groups. Subgroup analysis revealed the patients aged 65 or older had slight cognitive improvement, whereas younger patients demonstrated worsening of cognition after 12-month forniceal DBS treatment [103]. A two-year follow-up of the study reported the same conclusion. The forniceal DBS treatment had possible benefits in patients aged 65 and older [104]. A larger phase III multicenter RCT of forniceal DBS is currently underway to evaluate the effectiveness of cognition in the elderly with mild AD [105].

In addition to the fornix, the nucleus basalis of Meynert (NBM) is also considered a targeted region of DBS. A small-size, double-blind, sham-controlled phase I trial of NBM-DBS was assessed in patients with mild to moderate AD. At a 12-month follow-up assessment, two thirds of the patients had improved or stabilized cognitive performance [106]. The patients who responded to NBM-DBS had the characteristics of a higher baseline cognitive function and less advanced cortical atrophy [261,262].

#### 2.5.2. Vagus Nerve Stimulation

Vagus nerve stimulation is categorized into invasive and non-invasive methods. With invasive VNS (iVNS), the surgeon places the electrode at the left side of the tenth cranial nerve. The electrical stimulation is generated by a connected implanted pulse generator. Invasive VNS is approved for treating epilepsy and treatment-resistant depression. In 2002, a small-size pilot study of iVNS was researched in AD patients for 6 months. After a 3-month treatment, an estimated 90% of patients showed an improved MMSE performance, and 70% of the patients had better performance on ADAS-Cog. After 6 months of treatment, the response rate was maintained at 70% in AD patients [107]. The same researchers recruited more AD patients and followed them up for 12 months. After 12-month iVNS treatment, an estimated 70% of patients showed stabilized or improved cognitive performance [108].

Non-invasive VNS (nVNS) is a non-invasive intervention. The portable nVNS device provides electrical stimulation transcutaneously at the ear or neck and indirectly stimulates the auricular branch of the vagus nerve. Non-invasive VNS has been proven to be effective in the treatment of cluster headaches and migraines [263]. In the cognitively healthy elderly, a single session of nVNS improved associative memory in the treated group compared to the placebo. No serious or long-term adverse effects were reported [264]. The potential of nVNS in AD treatment has been suggested, but the clinical evidence is still lacking [265]. One double-blind sham-controlled crossover study is underway to evaluate the therapeutic effect of nVNS in patients with MCI [109].

#### 2.5.3. Transcranial Magnetic Stimulation

Transcranial magnetic stimulation (TMS) is a non-invasive brain stimulation technique. The TMS device produces an electric current through a coil wire, which is encased in plastic and placed above the patient’s scalp. The process produces a magnetic field across the cranial tissue and results in electrical stimulation at targeted sites of the brain [266]. The pulse of TMS can be single or repeated. Compared to single-pulse TMS, repetitive TMS (rTMS) modulates the cortical activity and promotes after effects beyond the stimulation period. Different rTMS protocols lead to variant after effects in the brain, with an inhibitory effect at low-frequency stimulation (≤1 Hz), and excitatory effects at high-frequency stimulation (≥5 Hz) [267]. A high-frequency rTMS protocol at the left dorsolateral prefrontal cortex (DLPFC) has been approved for treatment-resistant depression therapy in the United States [268].

High-frequency rTMS at the left DLPFC was also investigated in AD treatment. Three double-blind RCTs of 10 Hz rTMS were evaluated in MCI patients. Two studies of 2-week 10 Hz rTMS treatment showed a significant improvement in executive function in the treated group compared to the sham group [110,269]. One study of 4-week 10 Hz rTMS for evaluation in MCI patients is still recruiting and continuing [111].

The 20 Hz rTMS at the left DLPFC method was researched in studies for AD treatment. In AD patients, 2-week 20 Hz rTMS treatment led to improved language performance, and 4-week intervention brought an even greater change and longer-lasting effect [112]. The 20 Hz rTMS protocol was tested in AD patients with BPSD. Compared to the control group, which received low-dose antipsychotic medications alone, combined 20 Hz rTMS and medication resulted in significant improvement in both cognitive functions and BPSD symptoms after a 4-week treatment [113]. One double-blind RCT of 20 Hz rTMS was tested in patients with mild to moderate AD patients. The study aimed to evaluate the add-on effect of rTMS. All the participants received face–name associative memory cognitive training. After 4 weeks of treatment, the treated group showed better performance in trained associative memory than the sham group. Combined rTMS and cognitive training showed a greater benefit than cognitive training alone. The additional improvement was greater in participants with higher educational levels and cognitive baseline [114]. Another trial on the add-on effect of high-frequency rTMS was also investigated in AD treatment. Mild to moderate AD patients undergoing cognitive training received real or sham rTMS treatment for 4 weeks. Compared to the sham group, the treated group showed better performance in general cognitive and behavioral functions [115]. Stimulation at the left DLPFC seems to be the most popular and promising protocol in AD treatment.

Some studies of high-frequency rTMS targeted bilateral DLPFCs. The first study applied 20 Hz rTMS in patients with different degrees of AD. Following 5-day stimulation, patients receiving high-frequency rTMS showed cognitive and functional improvement in mild to moderate AD when compared to the sham group [117]. Another pilot crossover study of 4-week 20 Hz rTMS was evaluated in mild to moderate AD patients. The results revealed a stronger improvement in general cognition during the treatment phase than the sham phase [118]. Few studies have focused on the 5 Hz rTMS protocol in AD therapy. One clinical trial of 5 Hz rTMS compared the efficacy of different protocols in AD patients: simple (stimulation at left DLPFC) versus complex (stimulation at six other cortical sites). The results suggested that a 3-week intervention promoted both cognitive and functional improvement in both groups, and that there was no difference between the simple and complex protocols [116].

High-frequency rTMS is also performed at different targeted sites in the brain, such as the inferior frontal gyrus (IFG), superior temporal gyrus (STG), and parietal and posterior temporal lobes. One crossover RCT evaluated the clinical benefit of 10 Hz rTMS at the right IFG in MCI patients. All the patients received two sessions of stimulation in a random order: right IFG (active site), and right vertex (control site). The results indicated that high-frequency stimulation at the right IFG enhanced the improvement in attention and psychomotor speed, while the stimulation of the vertex showed a significant cognitive change [119]. Another crossover study tested the efficacy of high-frequency rTMS in patients with MCI and mild to moderate AD. Dementia patients had a pattern of gray matter atrophy, especially at the bilateral IFG, putamen, and cerebellum. The stimulation lasted a total of three sessions at 10 Hz over three regions in a random order: right IFG (active site), right STG (active site), and right vertex (control site). The stimulation of the right IFG and right STG revealed cognitive benefits, especially in attention and psychomotor speed performance. Patients with a greater gray matter volume reduction gained more benefit from the rTMS intervention [120]. The efficacy of high-frequency rTMS of the left parietal lobe (precuneus) was tested in MCI patients. The 20 Hz rTMS protocol was conducted for 2 weeks. The stimulation of the left parietal lobe revealed greater clinical benefits in episodic memory in the treated group than in the sham group, but the effect was not noted in other cognitive domains. Analysis of rTMS combined with electroencephalography uncovered the phenomenon of modulation of brain connectivity [121]. One double-blind RCT of 20 Hz rTMS targeted the region of bilateral posterior temporal regions of the brains of AD individuals. The study included mild to moderate AD patients and performed 6-week 20 Hz stimulation in the treated group and sham stimulation in the control group. The results showed that rTMS had advantages in the treatment of mild AD, with better performance in memory and language in the treated group than the sham group. However, the cognitive benefit was minimal or insignificant in moderate AD patients [122].

The low-frequency rTMS protocol has been less researched than the high-frequency protocol with regard to AD treatment. One randomized sham-controlled trial of 1 Hz rTMS at DLPFCs was applied in healthy individuals and MCI patients. The participants received two sessions on the same day, either at the right or left DLPFC. The results indicated that low-frequency rTMS enhanced recognition memory in both the healthy and MCI groups. Inhibition of the right DLPFC may modulate the excitability of the contralateral hemisphere [123]. In AD, one clinical study compared the efficacy of high-frequency (20 Hz) and low-frequency (1 Hz) rTMS targeting bilateral DLPFCs. The therapeutic efficacy of low-frequency rTMS was demonstrated to be less effective than high-frequency rTMS after a 5-day intervention [117].

A recent meta-analysis concluded that both high-frequency and low-frequency rTMS resulted in cognitive improvement in AD patients, with medium to large effect sizes [270]. The after effect of five or more sessions of rTMS could last from a few weeks to 4 months [114,115,116,117,118,270,271].

#### 2.5.4. Transcranial Electrical Stimulation

Transcranial electrical stimulation involves passing a weak electrical current (1–2 mA) among two or more electrodes on the subject’s scalp. The electrode positioning could be based on the international 10–20 electrode placement system, the neuronavigation system, or physiology-based placement. Transcranial electrical stimulation comes in two major forms: transcranial direct current stimulation (tDCS) and transcranial alternating current stimulation (tACS). In tDCS, the electrodes are divided into anodal or cathodal sites, while in tACS, the electrodes are active or reference sites. The applied electrical current is direct in tDCS but sinusoidal in tACS [272,273].

Transcranial direct current stimulation is the most common choice of transcranial electrical stimulation in treating AD. Studies into tDCS in AD have focused on several targeted regions: left DLPFC, left temporal lobe, and temporoparietal lobe [274]. In 2009, a study of anodal tDCS at the left DLPFC was first performed in patients with mild to moderate AD. The participants received true stimulation with an intensity of 2 mA for 30 min, and sham stimulation was conventionally set as 30 s. The results indicated that a single session of left DLPFC stimulation led to improved recognition memory [124]. The 2 mA tDCS protocol over the left DLPFC was tested in AD patients with repeated sessions. One double-blind RCT of tDCS included mild to moderate AD patients. The participants were classified into anodal tDCS, cathodal tDCS, and sham tDCS groups. The true stimulation was 25 min long. After 10 sessions, the active treatment group showed a higher MMSE score than the sham group, and the cognitive benefit was similar in the anodal and cathodal tDCS groups [125]. One double-blind phase II RCT of the 2 mA anodal tDCS protocol was tested in moderate AD patients with apathy. Each true session persisted for 20 min, and the total session number was six times. However, the intervention of tDCS showed no significant effect regarding cognitive, behavioral, or apathy symptoms in moderate AD. The study suggested that more than six sessions may promote clinical change in patients with moderate AD [126]. One recent study of at-home tDCS was applied in patients with early AD. The 2 mA tDCS protocol was applied daily for 6 months with 30-min sessions. Compared to the sham group, the treated group showed improved or stabilized cognition, with improvement in global and language functions and a decreased reduction in executive function [127].

In MCI patients, the anodal tDCS protocol showed clinical benefits in cognition. One double-blind RCT of 1.5 mA anodal tDCS was evaluated in MCI treatment. Each true session was 15 min, while sham stimulation was 10 s each session. This study indicated that a single session of tDCS enhanced the free recall and recognition of memory in the treated group compared to the sham group [128]. One pilot study compared the efficacy between 2 mA anodal tDCS and cognitive stimulation in MCI patients. The study also aimed to determine the optimal frequency of tDCS in MCI treatment. Each true session lasted for 30 min, with a variance of one to five sessions in the treatment phase. The results revealed that tDCS treatment resulted in a significant but mild improvement in some cognitive aspects, especially in selective attention, processing speed, and planning ability tasks. The optimal frequency of tDCS was three sessions per week. The conclusion should be warranted due to the session’s variability [129].

Several studies of tDCS targeted the region of the left parietal lobe. In 2009, the anodal tDCS at the left parietal region was tested in patients with mild to moderate AD. Single-session 30-min 2 mA anodal tDCS promoted superior improved recognition memory in the treated group compared to the sham group [124]. One study of 2 mA anodal tDCS over the left temporal area was assessed in patients with mild to moderate AD. The participants received six 30-min sessions. Active tDCS stimulation did not result in a significant change in verbal memory function [130].

The other targeted site of tDCS was the bilateral temporoparietal lobe. In 2008, the 1.5 mA tDCS protocol of the bilateral temporoparietal region was first tested in patients with mild AD. All participants received one sham stimulation and two true stimulations (anodal and cantonal) in a random order. Each true session was performed for 15 min, while the sham stimulation lasted only 10 s. The subgroup analysis showed that the anodal tDCS group gained improved word recognition, but the cathodal tDCS group experienced cognitive worsening instead [131]. One trial of 2 mA anodal tDCS at the bilateral temporal areas was evaluated in patients with mild to moderate AD. The participants received five 30-min sessions. The anodal stimulation enhanced the visual recognition memory in the treated group compared to the sham group [132]. Another trial of 2 mA anodal tDCS over the bilateral temporal areas was investigated with regard to cognitive and biological changes in patients with early AD. The patients received ten 20-min sessions. Significantly improved cognitive performance and increased total serum Aβ levels were observed in the treated group, but there was no change in tau or lipid peroxidase [133]. One recent study assessed the short-term and long-term effects of 2 mA tDCS over the left temporoparietal region in the treatment of advanced AD. The true stimulation was administrated for 20 min daily, for a total of 10 times. The sham stimulation was 10 s each time. At one month, the tDCS intervention stabilized the neuropsychological performance in the treated group, while the sham group showed a significant decline. The treated group continued with the frequency of tDCS for five sessions per month for 8 months. The protective effect of tDCS was maintained in long-term follow-up [134]. The after effect of tDCS generally lasted for at least 4 weeks [128,132,134].

tACS is a choice among transcranial brain stimulations. Some small trials have indicated that tACS may enhance specific cognitive functions in cognitively healthy populations [275,276]. The evidence of the therapeutic effect of tACS in AD patients is limited. In 2020, one pilot study of tACS over the left DLPFC was conducted in patients with MCI and mild to moderate AD. The study aimed to investigate the additional cognitive effect of combined tDCS in patients undergoing brain exercises. The stimulation was scheduled as sinusoidal waveforms at the frequency of 40 Hz with an intensity of 1.5 mA, from −0.75 to +0.75 mA. The treated group received two 30-min sessions per day, with 40 sessions in total. At a 4-week follow-up assessment, the tACS group showed a slight improvement in cognitive performance, while the non-tACS group demonstrated slight cognitive decline instead. The study showed that tACS had potential in AD treatment, and that the after effect may be maintained for 4 weeks [135]. One crossover RCT of 40 Hz tACS at 3 mA in AD treatment has been completed, in which the targeted region of tACS was the superior parietal cortex. The study was completed, but the results have not been published [136]. Several clinical trials of 40 Hz tACS are underway in AD treatment [137,277].

## 3. Discussion

In this review, we described the development of AD therapy during the past two decades, identifying five mainstream categories: anti-Aβ therapy, anti-tau therapy, anti-neuroinflammatory therapy, neuroprotective agents, and brain stimulation. Initially, the pathological markers Aβ and tau were the main targets of therapy. Immunotherapies became the most popular method among the two fields. The target of intervention gradually shifted from specific pathological markers to complex mechanisms, such as neuroinflammatory and neurodegenerative processes. Compared to the pharmacological field, non-pharmacological interventions went even further. Among brain stimulation approaches, non-invasive methods have been more tolerable than invasive techniques, and the most commonly studied methods were rTMS and tDCS. rTMS and tDCS showed convincing outcomes in cognitive enhancement and maintenance. Non-invasive brain stimulations have the potential to be the next trend in AD treatment.

Drug repositioning is another potential method for accelerating pharmacological development. Drug repositioning has many advantages. First, repositioning existing drugs to new therapeutic uses is less expensive than developing a new drug. Second, both the safety and tolerance of existing drugs have already been investigated. It is easier for these drugs to achieve advanced clinical stages to evaluate the therapeutic effect in AD. Some potential drugs were proposed in this way, such as lithium, metformin, levetiracetam, and sodium benzoate. The challenges lie in the choice of existing drugs, which depends on our understanding of AD pathogenesis.

Precision medicine is another issue. Some patients gained more clinical benefits from a specific intervention. For example, sodium benzoate showed more cognitive improvement in female patients than in males. Phospholipid DHA supplements showed a preventive effect on AD in APOE4 carriers. Most of the AD patients who responded to rTMS and tDCS were in the early stage of AD. The etiology of AD is considered multifactorial, which may be distinct in each individual. Choosing the appropriate interventions according to the characteristics of AD patients can help to achieve a therapeutic effect as soon as possible and minimize the harm and adverse effects of treatment. Designing personalized interventions is one of the most critical milestones of further AD treatment.

## 4. Future Research Direction

In pharmacological interventions, researching the potential agents is still a challenge. AD is a multifactorial disorder and involves several pathogenic mechanisms: misfolded protein aggregation, neuroinflammatory process, neurodegeneration, and insulin dysregulation. Drug repositioning is a possible effective method. The potential candidates include anti-inflammatory agents, neuroprotective agents, and antidiabetic agents. As biophysical and structural biology experimental approaches progress, the pathophysiological mechanisms of Aβ and tau are being uncovered. The knowledge of AD pathogenesis helps us to find potential compounds or to design further immunotherapies.

In non-pharmacological interventions, standardizing the settings of the protocol is the current challenge. For example, to determine the protocol of rTMS, the parameters include the targeted sites, frequency, duration of each session, and schedule. The same rTMS protocol may show inconsistent efficacy in patients at different stages of AD. Designing several standardized protocols according to the disease severity is a possible strategy in the future.

Selecting responsive subgroups is important in both pharmacological and non-pharmacological interventions. The characteristics of patients are involved in designing the treatment, including age, gender, genetic factors, medical diseases, environmental factors, and lifestyles. Advances in machine learning allow us to deal with complex factors and build models which predict the optimal therapeutic regimens for AD patients. Further research is required to uncover the relationship between patients’ characteristics and response to a specific treatment. With further research efforts, the practice of precision medicine is possible and anticipated.

## Figures and Tables

**Figure 1 ijms-22-08208-f001:**
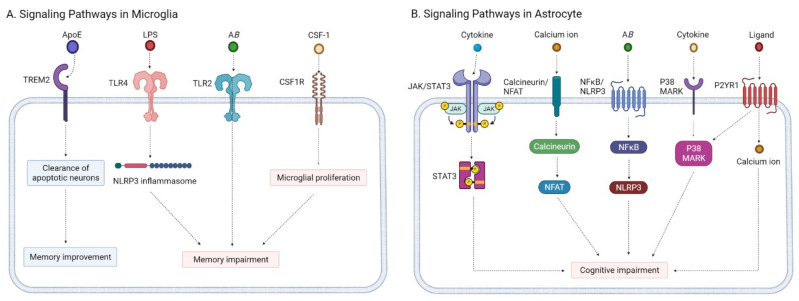
Signaling pathways of microglia modulators and astrocyte modulators. (**A**) Signaling pathways in microglia; (**B**) signaling pathways in astrocytes. Created with BioRender.com. * TREM2—triggering the receptor expressed on myeloid cells 2, TLR—Toll-like receptor, CSF1R—colony-stimulating factor-1 receptor, JAK—Janus kinase, STAT3—signal transducer and activator of transcription 3, NFAT—nuclear factor of activated T cells, NFκB—nuclear factor-kB, NLRP3—nod-like receptor family pyrin domain containing 3, MAPK—mitogen-activated protein kinase, P2Y1R—P2Y1 purinoreceptor.

**Figure 2 ijms-22-08208-f002:**
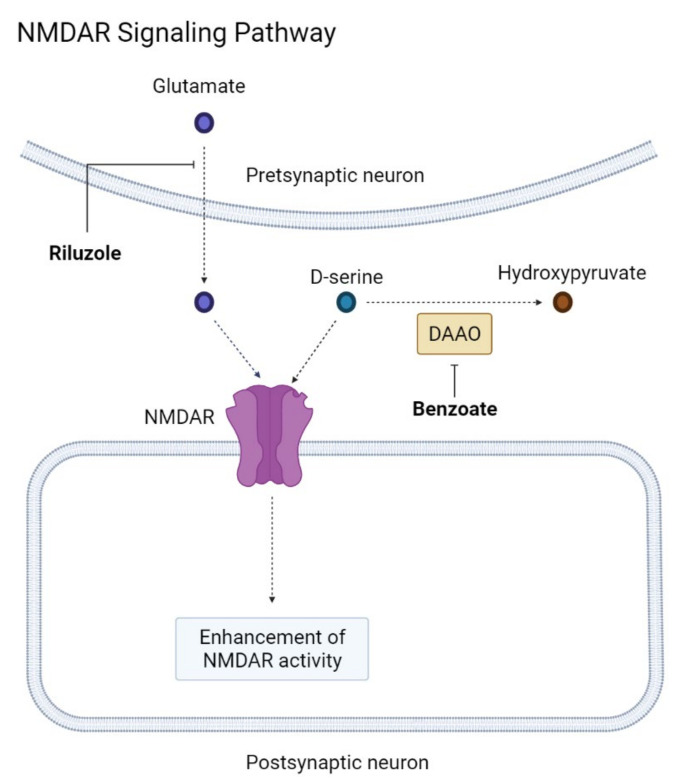
NMDAR signaling pathway. Created with BioRender.com. NMDAR—N-methyl-D-aspartate receptor, DAAO—D-amino acid oxidase. The sharo arrow means activation of the chemical reaction. The blunt head arrow means inhibition of the chemical reaction.

**Figure 3 ijms-22-08208-f003:**
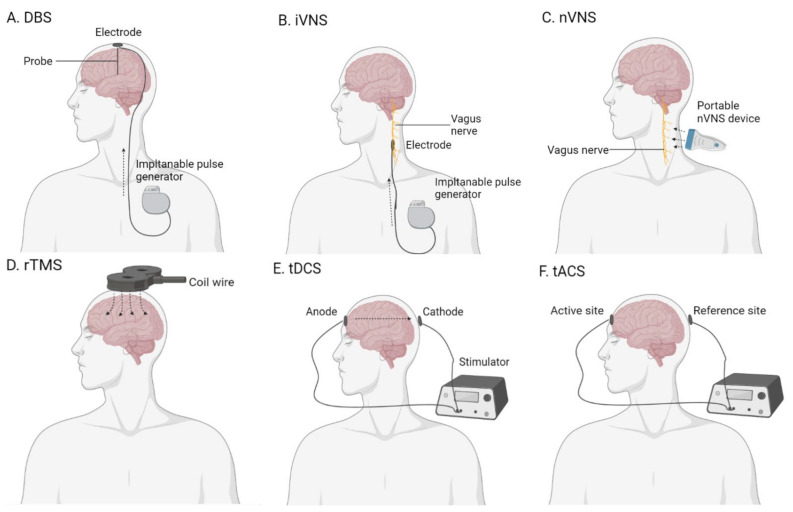
The diagram of the brain stimulation devices. (**A**) Deep-brain stimulation (DBS); (**B**) invasive vagus nerve stimulation (iVNS); (**C**) non-invasive vagus nerve stimulation (nVNS); (**D**) repetitive transcranial magnetic stimulation (rTMS); (**E**) transcranial direct current stimulation (tDCS); (**F**) transcranial alternating current stimulation (tACS). Created with BioRender.com. The sharp arrow means the direction of energy flow.

**Table 1 ijms-22-08208-t001:** Summary of pharmacological interventions against AD.

Class of Drugs	Compounds	Mechanism	Subjects	Status	Summary	[Ref]
1. Anti-amyloid therapy						
Secretase inhibitor	Verubecestat	BACE1 inhibitor	Prodromal to moderate AD	Phase II/III	Lack of efficacy	[20,21]
	Atabecestat	BACE1 inhibitor	Prodromal AD	Phase II/III	Cognitive worsening, psychiatric disorder	[22]
	Lanabecestat	BACE1 inhibitor	MCI to mild AD	Phase III	Cognitive worsening, weight loss, psychiatric disorder	[23]
	LY3202626	BACE1 inhibitor	Mild AD	Phase III	Lack of efficacy	[24]
	Umibecestat	BACE1 inhibitor	Cognitively healthy APOE4 carriers	Phase II/III	Completed, failed analysis due to small number of events	[25]
	Elenbecestat	BACE1 inhibitor	MCI to moderate AD	Phase III	Lack of efficacy, nightmare	[26,27]
	Semagacestat	γ-secretase inhibitor	Mild to moderate AD	Phase III	Lack of efficacy, skin cancer, weight loss, hematologic disorder, infection	[28]
	Avagacestat	γ-secretase inhibitor	MCI	Phase II	Lack of efficacy, non-melanoma cancer, gastrointestinal symptoms	[29]
	Tarenflurbil	γ-secretase modulator	Mild AD	Phase II	Lack of efficacy, anemia, infection	[30]
Aβ aggregation inhibitor	PBT1	MPAC	MCI to moderate AD	Phase II	Rescue of cognitive decline in severely affected patients (ADAS-cog ≥25), visual impairment	[31]
	PBT2	MPAC	Mild to moderate AD	Phase II	Lack of efficacy, large individual variance	[32,33]
Aβ immunotherapy	ACI-24	Aβ vaccine	Adults with Down syndrome	Phase II	Lack of immunogenicity	[34]
	CAD106	Aβ vaccine	Mild AD	Phase II	Lack of efficacy	[34]
	UB-311	Aβ vaccine	Mild AD	Phase II	No published data	[34]
	ABVac40	Aβ vaccine	MCI to mild AD	Phase II	Ongoing	[34]
	BAN2401	Monoclonal antibody	MCI to mild AD	Phase III	Modest efficacy among APOE4 carriers	[35]
	Gantenerumab	Monoclonal antibody	Prodromal to mild AD	Phase III	Lack of efficacy	[36]
	Aducanumab	Monoclonal antibody	Monoclonal antibody	Phase III	Termination, little change in efficacy FDA approval for now	[37,38]
2. Anti-tau therapy						
Phosphatase modifier	Selenate	PP2A activator	Mild to moderate AD	Phase II	Lack of efficacy	[39,40]
Kinase inhibitor	Roscovitine	CDK5 inhibitor	5XFAD mice	In vivo	Prevention of tau phosphorylation	[41,42]
	Flavopiridol	CDK5 inhibitor	CD1 mice	In vivo	Rescue of cognitive decline	[41,42]
	Tideglusib	GSK3β inhibitor	Mild to moderate AD	Phase II	Lack of efficacy, transaminase increase	[43]
	Lithium	GSK3β inhibitor	MCI	Phase II	Rescue of cognitive decline	[44,45,46]
Tau aggregation inhibitor	MB	Disrupts polymerization	Mild to moderate AD	Phase II	Cognitive improvement	[47]
	LMTX	Disrupts polymerization	Mild to moderate AD	Phase III	Lack of efficacy	[48]
	Curcumin	Decreases β-sheet formation in tau	Cognitively healthy elderly	Phase II	Improvement in working memory (short-term course)	[49,50]
Microtubule stabilizer	EpoD	Enhances microtubule bundling	Mild AD	Phase I	Discontinuation, frequent adverse effects without published data	[51]
	NAP	Protects microtubules from katanin disruption	MCI	Phase II	Cognitive and functional improvement	[52,53]
	TPI-287	Stabilizes microtubules	Mild to moderate AD	Phase I	Rescue of cognitive decline, anaphylactoid reactions	[54]
Tau immunotherapy	AADvac1	Tau vaccine	Mild AD	Phase II	Completed, no published data	[55]
	ACI-35	Tau vaccine	Mild to moderate AD	Phase I	Safe and tolerated	[56]
	Aβ 3–10-KLH	Tau vaccine	3×Tg-AD mice	In vivo	Cognitive improvement	[57]
	BIIB092	Monoclonal antibody	Early AD	Phase II	Ongoing	[58]
	ABBV-8E12	Monoclonal antibody	Early AD	Phase II	Ongoing	[59,60]
	RO7105705	Monoclonal antibody	Prodromal to moderate AD	Phase II	Ongoing	[61,62]
	BIIB076	Monoclonal antibody	Healthy volunteers, MCI	Phase I	Safe and tolerated	[63]
	LY3303560	Monoclonal antibody	Early AD	Phase II	Completed, no available data	[64]
	JNJ-63733657	Monoclonal antibody	Early AD	Phase II	Ongoing	[65]
	UCB0107	Monoclonal antibody	Healthy volunteers	Phase I	Ongoing	[66,67]
3. Anti-neuroinflammatory therapy						
Microglia modulator	Thymoquinone	TLR4 inhibitor	AD mice induced by AlCl_3_	In vivo	Rescue of cognitive impairment	[68]
	Ethyl pyruvate	TLR4 inhibitor	AD mice induced by AlCl_3_	In vivo	Rescue of cognitive impairment	[68]
	TAK-242	TLR4 inhibitor	APP/PS1 mice	In vivo	Cognitive improvement	[68]
	GW2580	CSF1R inhibitor	APP/PS1 mice	In vivo	Recovery of short-term memory and behavioral deficit	[69]
	JN-J527	CSF1R inhibitor	P301S mice	In vivo	Functional improvement	[70]
	PLX3397	CSF1R inhibitor	5XFAD mice	In vivo	Recovery of spatial and emotional memory deficit	[71]
Astrocyte modulator	Stattic	STAT3 inhibitor	5XFAD mice	In vivo	Rescue of learning and memory impairment	[72,73]
	FK506	Calcineurin/NFAT inhibitor	MCI to AD	Phase II	Not yet recruiting	[74]
	SB202190	P38 MAPK inhibitor	Wip1-deficient mice	In vivo	Rescue of learning and memory impairment	[75]
	PD169316	P38 MAPK inhibitor	Aβ-injected mice	In vivo	Rescue of spatial memory and learning impairment	[75]
	MW108	P38 MAPK inhibitor	hTau mice	In vivo	Rescue of cognitive impairment	[76]
	NJK14047	P38 MAPK inhibitor	5XFAD mice	In vivo	Cognitive improvement	[77]
	MRS2179	P2Y1R inhibitor	APPPS1 mice	In vivo	Spatial learning improvement	[78]
	BPTU	P2Y1R inhibitor	APPPS1 mice	In vivo	Spatial learning improvement	[78]
Insulin resistance management	Intranasal insulin therapy	Intranasal supplement	MCI to moderate AD	Phase II	Cognitive improvement, modulation by APOE4 genotype	[79,80]
			MCI to AD	Phase II/III	Lack of efficacy	[81]
	Liraglutide	Incretin receptor agonist	Mild AD	Phase II	Delay of cognitive impairment	[82,83]
	Metformin	Biguanide	MCI	Phase II	Reduction in recall memory decline	[84]
			MCI to early AD	Phase II	Executive functional improvement	[85]
	Gemfibrozil	PPAR-α agonist	MCI	Phase I	Completed, no published data	[86]
	Pioglitazone	PPAR-γ agonist	Mild AD	Phase II	Cognitive improvement	[87]
			MCI	Phase III	Lack of efficacy	[88,89]
	T3D-959	Hybrid PPAR-δ/γ agonist	STZ-induced AD mice	In vivo	Reduction in neuroinflammation	[90]
Microbiome therapy	Sodium oligomannate	Dysbiosis of gut microbiota	Mild to moderate AD	Phase III	Cognitive improvement	[91,92]
4. Neuroprotective agents						
Antiepileptic drug	Levetiracetam	SV2A receptor	MCI	Phase III	Ongoing	[93]
	Gabapentin	VGCCs inhibitor	Moderate to severe AD	Phase IV	Ongoing	[94]
NMDAR modification	Sodium benzoate	DAAO inhibitor	MCI to mild AD	Phase II	Cognitive improvement	[95]
			MCI	Phase II	Cognitive and functional improvement	[96]
			Moderate to severe AD with BPSD	Phase II	Cognitive benefit in female gender	[97]
	Riluzole	Glutamate modulator	Mild AD	Phase II	Completed, no published data	[98]
	Troriruzole	Glutamate modulator	Mild to moderate AD	Phase II	Ongoing	[99]
Omega 3 polyunsaturated fatty acid supplements	DHA	Anti-oxidative effect	Mild to moderate AD	Phase III	Lack of efficacy	[100]
			Cognitively healthy elderly	Phase II	Ongoing	[101]
	Icosapent ethyl	Anti-oxidative effect	Cognitively healthy elderly	Phase III	Ongoing	[102]

BACE1—β-secretase1, APOE4—apolipoprotein E type 4, PBT1—clioquinol, PBT2—second-generation clioquinol, MPAC—metal protein attenuating compound, ADAS-cog—Alzheimer’s Disease Assessment Scale–Cognitive Subscale, MB—methylene blue, EpoD—Epothilone D, NAP—davunetide, TPI-287—abeotaxane, DHA—docosahexaenoic acid.

**Table 2 ijms-22-08208-t002:** Summary of non-pharmacological interventions against AD.

Methods	Targeted Region	Protocol	Subjects	Status	Summary	[Ref]
1. Deep-brain stimulation						
DBS	Fornix	Forneceal DBS	Mild AD	Phase II	Slight cognitive benefit in the elderly	[103,104]
			Mild AD	Phase III	Ongoing	[105]
	NBM	NBM-DBS	Mild to moderate AD	Phase I	Cognitive stabilization and improvement, response rate 67%	[106]
2. Vagus nerve stimulation						
VNS	Tenth cranial nerve	Invasive VNS	Probable AD	Phase I	Cognitive stabilization and improvement, response rate 70%	[107,108]
	Tenth cranial nerve	Non-invasive VNS	MCI	Not Applicable	Ongoing	[109]
3. Transcranial magnetic stimulation						
High-frequency rTMS	Left DLPFC	10 Hz/120% MT/3000 pulses per session/10 sessions/2 weeks *	MCI	Phase IV	Executive functional improvement	[110]
		10 Hz/120% MT/2000 pulses per session/20 sessions/4 weeks *	MCI	Not Applicable	Ongoing	[111]
		20 Hz/100% MT/2000 pulses per session/20 sessions/4 weeks *	Moderate AD	Not Applicable	Improved language performance	[112]
		20 Hz/80% MT/1200 pulses per session/20 sessions/4 weeks *	AD patients with BPSD	Not Applicable	Cognitive and functional improvement	[113]
		20 Hz/100% MT/2000 pulses per session/20 sessions/4 weeks *	Mild to moderate AD	Not Applicable	Improvement in trained associative memory, add-on effect	[114]
		20 Hz/80–100% MT/1000 pulses per session/20 sessions/4 weeks *	Mild to moderate AD	Not Applicable	Cognitive and functional improvement, add-on effect	[115]
		5 Hz/100% MT/1500 pulses per session/15 sessions/3 weeks *	Probable AD	Not Applicable	Cognitive and functional improvement	[116]
	Bilateral DLPFCs	20 Hz/90% MT/2000 pulses per session/5 sessions/5 days *	Mild to severe AD	Not Applicable	Cognitive and functional improvement in mild to moderate AD	[117]
		20 Hz/90–100% MT/2000 pulses per session/13 sessions/4 weeks *	Mild to moderate AD	Not Applicable	Cognitive improvement	[118]
	Right IFG	10 Hz/90% MT/2250 pulses per session/single session *	MCI	Not Applicable	Improvement in attention and psychomotor speed	[119]
		10 Hz/90% MT/2250 pulses per session/single session *	MCI to moderate AD	Not Applicable	Cognitive improvement	[120]
	Right STG	10 Hz/90% MT/2250 pulses per session/single session *	MCI to moderate AD	Not Applicable	Cognitive improvement	[120]
	Left parietal lobe	20 Hz/100% MT/1600 pulses per session/10 sessions/1 week *	Early AD	Not Applicable	Improvement in episodic memory	[121]
	Bilateral parietal lobes	20 Hz/Unavailable MT/1 h per session/30 sessions/6 weeks *	Mild to moderate AD	Not Applicable	Better performance in memory and language in mild AD	[122]
Low-frequency rTMS	Bilateral DLPFCs	1 Hz/90% MT/600 pulses per session/2 sessions/1 day *	Healthy individuals-MCI	Not Applicable	Improvement in recognition memory	[123]
		1 Hz/100% MT/2000 pulses per session/5 sessions/5 days *	Mild to severe AD	Not Applicable	Less cognitive efficacy than high-frequency rTMS	[117]
4. Transcranial electrical stimulation						
Transcranial direct current stimulation	Left DLPFC	2 mA/30 min per session/single session **	Mild to moderate AD	Not Applicable	Improved recognition memory	[124]
		2 mA/25 min per session/10 sessions/2 weeks **	Mild to moderate AD	Not Applicable	Cognitive improvement	[125]
		2 mA/20 min per session/6 sessions/2 weeks **	Moderate AD	Phase II	No cognitive or behavioral improvement, no change in apathy symptoms Requirement of more than 6 sessions	[126]
		2 mA/30 min per session/daily session/6 months **	Early AD	Not Applicable	Cognitive and functional improvement, rescue of executive function	[127]
		1.5 mA/15 min per session/single session **	MCI	Not Applicable	Enhanced free recall and recognition of memory	[128]
		2 mA/30 min per session/1-5 sessions **	MCI	Not Applicable	Improvement in selective attention, processing speed, and planning ability tasks Optimal frequency of 3 sessions/week	[129]
	Left parietal lobe	2 mA/30 min per session/single session **	Mild to moderate AD	Not Applicable	Improvement in recognition memory	[124]
		2 mA/30 min per session/6 sessions/10 days **	Mild to moderate AD	Not Applicable	No improved verbal memory function	[130]
	Bilateral temporoparietal lobe	1.5 mA/15 min per minute/2 sessions (anodal and cathodal) **	Mild AD	Not Applicable	Improved word recognition in anodal group, cognitive worsening in cathodal group	[131]
		2 mA/30 min per session/5 sessions/1 week **	Mild to moderate AD	Not Applicable	Improvement in visual recognition memory	[132]
		2 mA/20 min per session/10 sessions/2 weeks **	Early AD	Not Applicable	Improved cognitive performance	[133]
	Left temporoparietal lobe	2 mA/20 min per session/10 sessions/2 weeks **	Advanced AD	Not Applicable	Stabilized neuropsychological performance, long-term protective effect	[134]
Transcranial alternating current stimulation	Left DLPFC	40 Hz/1.5 mA/30 min per session/40 sessions/4 weeks ***	MCI to moderate AD	Not Applicable	Improved cognitive performance	[135]
	Superior parietal cortex	40 Hz/3 mA/30 min per session/single session ***	AD patients	Not Applicable	Completed, no published data	[136]
	Left angular gyrus	40 Hz/unavailable intensity/20 min per session/3 sessions ***	Healthy individuals to mild AD	Not Applicable	Ongoing	[137]

NBM—nucleus basalis of Meynert, DLPFC—dorsolateral prefrontal cortex, MT—motor threshold. * Protocol of rTMS: Frequency/Intensity/Number of pulses per session/Total number of sessions/Duration. ** Protocol of tDCS: Current intensity/Stimulation duration/Total number of sessions/Duration. *** Protocol of tACS: Frequency/Intensity/Stimulation duration/Total number of sessions/Duration.

## Data Availability

Data is contained within the article.

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
