# Peer review of "Novel Therapeutic Approaches for Alzheimer’s Disease: An Updated Review"

_ijms, 2021, doi:10.3390/ijms22158208_

Round 1

Reviewer 1 Report

Reviewer Comments:

  1. At first, author should pay attention to keywords and abstract. Key words do not support the abstract.
  2. Line: 16-20 is not well explained or meaningless.
  3. Author should add a linking sentence before line: 41
  4. As authors summarized the pharmacological and non-pharmacological approaches for AD, it would be better to add the pathogenesis of AD before section 2. You may read these papers : https://www.nature.com/articles/s41583-018-0067-3; https://pubmed.ncbi.nlm.nih.gov/33610813/
  5. Authors do not describe this section (2.1.2. Aβ aggregation inhibitors) properly. There are several natural compounds which can significantly inhibit the Aβ aggregation. You may read these papers : https://www.nature.com/articles/aps201114; https://www.frontiersin.org/articles/10.3389/fnins.2020.619667/full
  6. There are so many paragraphs in section : 2.3.2. Please revise and rewrite.
  7. It would be better to add a figure through adding the subsection under section: 2.3
  8. Author should make a table to summarize the pharmacological and non-pharmacological interventions that are effective against AD (in vivo and clinical studies).
  9. I found it really nice that the authors mention a non-pharmacological approach, for example, brain stimulation. It would be more clear to readers if authors add a figure how brain stimulation might be a crucial option against AD.
  10. Please add a new section as “Future research direction” based on the current review.
  11. The overall writing is generally good, there are a number of grammatical and word-usage errors that impact on the clarity of the paper. This is at a level where an English language editor's input would be helpful.

Reviewer 2 Report

The review paper discusses an important and timely topic.

Some suggestions are provided to improve the scientific content of the paper.

  • Section "Novel therapeutic approach": authors should briefly discuss some of the reasons behind the lack of efficacy and/or toxicity of the discussed compounds.
  • In “Insulin resistance management”
  • About non-pharmacological strategies, it would be interesting to have a subsection devoted to physical activity (particularly the effects of aerobic exercise) and another subsection devoted to diets (e.g. Ketone-based diets).
  • The paper could be enriched by the inclusion of one or two figures summarizing the pharmacological and non-pharmacological strategies debated in the paper.

Round 2

Reviewer 1 Report

  1. Need to cite the reference in Line no: 67-69
  2. Line:86; “reducing the accumulation of Aβ” instead of “reducing the amyloid burden” and cite this article (https://pubmed.ncbi.nlm.nih.gov/16332141/)
  3. Line 106: cite this article (https://pubmed.ncbi.nlm.nih.gov/32464743/)
  4. Line 128-130: specify the name of the compounds
  5. Need to draw up a figure in section 2.5 including all subsections.
  6. Line: 714, Aβ instead of AB
  7. Line: 719-720 not clear
  8. Line 722: In many studies but no reference??? Revise this sentence
  9. Section 2.3; several signaling pathways are described in the text. It will make abstruse to the reader without drawing a figure.

Reviewer 2 Report

The paper has been improved.

Author Response

Thank you very much.